# Investigating Large Neighbourhood Search for Bus Driver Scheduling

**Primary Keywords:** *(1) Applications;*

## Abstract

The Bus Driver Scheduling Problem (BDSP) is a combinatorial optimisation problem with high practical relevance. The aim is to assign bus drivers to predetermined routes while minimising a specified objective function that considers operating costs as well as employee satisfaction. Since we must satisfy several rules from a collective agreement and European regulations, the BDSP is highly constrained. Hence, using exact methods to solve large real-life-based instances is computationally too expensive, while heuristic methods still have a considerable gap to the optimum. Our paper presents a Large Neighbourhood Search (LNS) approach to solve the BDSP. We propose several novel destroy operators and an approach using Column Generation to repair the sub-problem. We analyse the impact of the destroy and repair operators and investigate various possibilities to select them, including adaptivity. The proposed approach improves all the upper bounds for larger instances that exact methods cannot solve, as well as for some mid-sized instances, and outperforms existing heuristic approaches for this problem on all benchmark instances.

## 1 Introduction

The BDSP has evident practical relevance. Rules stem from different sources like EU regulations, national laws, or collective agreements, and can become very complicated. For real-life application, it is very important to consider not only paid working time, but also the overall well-being of the employees.

The BDSP can be seen as a highly complex part of the general transportation planning system, which includes Vehicle Scheduling, Crew Rostering, and Timetabling (Wren 2004). This problem is an **NP**-hard problem, even when only working time constraints are imposed (Fischetti, Martello, and Toth 1987). Different variants of BDSPs have been studied from the early 60's (Wren 2004; Wren and Rousseau 1995). Regarding exact methods, the BDSP is often modelled as a Set Partitioning Problem and Column Generation is often used (Smith and Wren 1988; Lin and Hsu 2016; Portugal, Lourenço, and Paixão 2008; Kletzander, Musliu, and Van Hentenryck 2021). However, due to the need to solve very large real-world problems in a reasonable time, several heuristic methods have been studied for BDSP: some examples are Greedy (Martello and Toth 1986), Tabu Search (Shen and Kwan 2001; Kletzander,

Mazzoli, and Musliu 2022), Simulated Annealing (Kletzander and Musliu 2020), GRASP (De Leone, Festa, and Marchitto 2011), CMSA (Rosati et al. 2023), and Genetic Algorithm (Li and Kwan 2003; Lourenço, Paixão, and Portugal 2001).

The constraints of the BDSP depend on the country's legal regulation, and in our case, they follow the Austrian *collective agreement for employees in private omnibus providers* (WKO 2019) using the rules for regional lines. In particular, the collective agreement has stringent rules requiring the drivers to take frequent breaks, with the eventual option of splitting them into multiple parts. This problem has been introduced recently in the literature, and to the best of our knowledge, the recently introduced exact approach based on Branch and Price (Kletzander, Musliu, and Van Hentenryck 2021), and meta-heuristic and hyper-heuristic based approaches (Kletzander and Musliu 2020; Kletzander, Mazzoli, and Musliu 2022; Kletzander and Musliu 2022; Rosati et al. 2023; Kletzander and Musliu 2023) represent the current state of the art for this problem. Although these approaches give very good results, exact methods are computationally too expensive for large real-life-based instances and heuristic methods cannot obtain optimal solutions. Therefore, the study of new methods to tackle this problem is an interesting research question with practical implications. In this paper, we present a novel approach based on the Large Neighbourhood Search framework, which has been successfully used for solving other challenging real-life problems. However, applying this method to our problem domain requires innovative ideas for the destroy and repair operators, as well as a detailed investigation into their impact and the choice of parameters. Our proposed approach significantly improves upon existing results, particularly for larger instances that cannot be solved with exact methods, and outperforms state-of-the-art heuristic approaches on all benchmark instances.

The main contributions of this paper are:

- Our paper presents a novel solution approach for the BDSP, based on Large Neighbourhood Search (LNS). The proposed method includes three innovative destroy operators and leverages a column-generation-based technique as repair operator.

- We explore different strategies for operator selection including an adaptive approach.

- We analyse the impact of destroy operators on the algorithm's performance and compare the use of Column Generation with full Branch and Price to repair the subproblems.

- We evaluate the performance of our LNS algorithm by comparing it with other state-of-the-art methods using existing benchmark instances from the literature. The results demonstrate that our LNS algorithm improves the upper bound for a significant number of instances, including some mid-sized instances and all large instances, and it outperforms all existing heuristic methods.

## 2 Problem Description

The investigated Bus Driver Scheduling Problem deals with the assignment of bus drivers to vehicles that already have a predetermined route for one day of operation. The problem specification is taken from Kletzander and Musliu (2020).

### 2.1 Problem Input

The bus routes are given as a set $L$ of individual bus legs, each leg $\ell \in L$ is associated with a tour $tour_\ell$ (corresponding to a particular vehicle), a start time $start_\ell$, an end time $end_\ell$, a starting position $startPos_\ell$, and an end position $endPos_\ell$. The actual driving time for the leg is denoted by $drive_\ell$. The given instances use $drive_\ell = end_\ell - start_\ell$.

The set $L$ is totally ordered by $start$, using $tour$ as tie-breaker.

| $\ell$ | $tour_\ell$ | $start_\ell$ | $end_\ell$ | $startPos_\ell$ | $endPos_\ell$ |
|---|---|---|---|---|---|
| 1 | 1 | 400 | 495 | 0 | 1 |
| 2 | 1 | 510 | 555 | 1 | 2 |
| 3 | 1 | 560 | 502 | 2 | 1 |
| 4 | 1 | 508 | 540 | 1 | 0 |

Table 1: A Bus Tour Example

Table 1 shows a short example of one particular bus tour. The vehicle starts at time $400$ (6:40 AM) at position 0, does multiple legs between positions 1 and 2 with waiting times in between and finally returns to position 0. A valid tour never has overlapping bus legs and consecutive bus legs satisfy $endPos_i = startPos_{i+1}$. A tour change occurs when a driver has an assignment of two consecutive bus legs $i$ and $j$ with $tour_i \neq tour_j$. A time distance matrix specifies, for each pair of positions $p$ and $q$, the time $d_{p,q}$ a driver takes to get from $p$ to $q$ when not actively driving a bus. If no transfer is possible, then $d_{p,q} = \infty$. $d_{p,q}$ with $p \neq q$ is called the *passive ride time*. $d_{p,p}$ represents the time it takes to switch tour at the same position, but is not considered passive ride time. Finally, each position $p$ is associated with an amount of working time for starting a shift ($startWork_p$) and ending a shift ($endWork_p$) at that position. The instances in this paper use $startWork_p = 15$ and $endWork_p = 10$ at the depot ($p = 0$). These values are 0 for other positions. For the given instances, the number of legs is proportional to the number of bus tours with approximately $n_{\text{legs}} \approx 10 \cdot n_{\text{tours}}$.

### 2.2 Solution

A solution $S$ to the problem is an assignment $S \colon L \to E$, where $E \subseteq \mathbb{N}$ is the set of employees. The number of drivers is not given, but one can imagine setting it as large as needed to have a feasible solution.

Equivalently, it is useful to represent a solution by a set of *shifts*, that is the work scheduled to be performed by a driver in one day (Wren 2004). More precisely, the shift of driver $e \in E$ is the preimage $L_e = S^{-1}(\{e\})$ with the total order induced by $L$. Hence, the notion of *consecutive* bus legs in a shift is well-defined.

Each shift of a driver $e \in E$ must be feasible according to the following criteria:

- No overlapping bus legs are assigned to $e$.
- Changing tour or position between consecutive legs $i, j \in L_e$ requires
$$start_j \geq end_i + d_{endPos_i, startPos_j}.$$
- The shift $L_e$ respects all hard constraints regarding work regulations as specified in the next section.

### 2.3 Work and Break Regulations

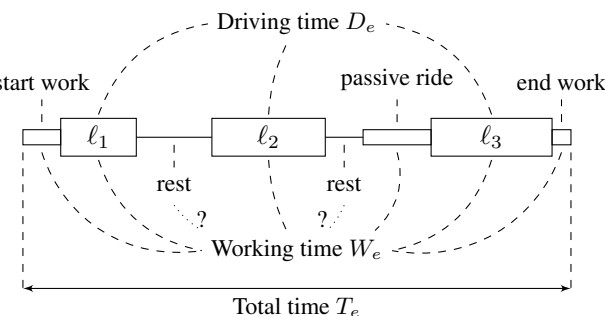

Figure 1: Example shift (Kletzander and Musliu 2020)

Valid shifts for drivers are constrained by work regulations and require frequent breaks. First, different measures of time related to an employee $e$ containing the set of bus legs $L_e$ need to be distinguished, as visualised in Figure 1:

- The total amount of driving time: $D_e = \sum_{i \in L_e} drive_i$.
- The span from the start of work until the end of work $T_e$ with a maximum of $T_{\max} = 14\,\text{h}$.
- The working time $W_e = T_e - unpaid_e$, which does not include certain unpaid breaks.

**Driving Time Regulations.** The maximum driving time is restricted to $D_{\max} = 9\,\text{h}$. The whole distance $start_j - end_i$ between consecutive bus legs $i$ and $j$ qualifies as a driving break, including passive ride time. Breaks from driving need to be taken repeatedly after at most 4 h of driving time. In case a break is split in several parts, all parts must occur before a driving block exceeds the 4 h limit. Once the required amount of break time is reached, a new driving block starts. The following options are possible:

- One break of at least 30 min;
- Two breaks of at least 20 min each;
- Three breaks of at least 15 min each.

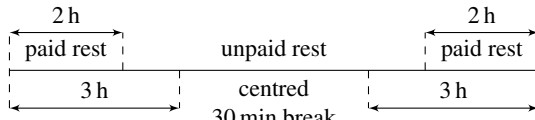

Figure 2: Rest break positioning (Kletzander and Musliu 2020)

**Working Time Regulations.** The working time $W_e$ cannot exceed 10 h and has a soft minimum of 6.5 h. If the employee is working for a shorter period of time, the difference has to be paid anyway.

A minimum rest break is required according to the following options:

- $W_e < 6$ h: no rest break;
- $6$ h $\leq W_e \leq 9$ h: at least 30 min;
- $W_e > 9$ h: at least 45 min.

The rest break may be split into one part of at least 30 min and one or more parts of at least 15 min. The first part has to occur after at most 6 h of working time. Whether rest breaks are paid or unpaid depends on break positions according to Figure 2. Every period of at least 15 min of consecutive rest break is unpaid as long as it does not intersect the first 2 or the last 2 hours of the shift (a longer rest break might be partially paid and partially unpaid). The maximum amount of unpaid rest is limited:

- If 30 consecutive minutes of rest break are located such that they do not intersect the first 3 h of the shift or the last 3 h of the shift, at most 1.5 h of unpaid rest are allowed;
- Otherwise, at most one hour of unpaid rest is allowed.

**Shift split.** If a rest break is at least 3 h long, it is instead considered a shift split, which is unpaid and does not count towards $W_e$. However, such splits are typically regarded badly by the drivers. A shift split counts as a driving break, but does not contribute to rest breaks.

## 2.4 Objective function

We minimise the objective function combining cost and employee satisfaction defined in previous work (Kletzander and Musliu 2020):

$$z = \sum_{e \in E} \left(2\,W'_e + T_e + ride_e + 30\,chg_e + 180\,split_e\right) \quad (1)$$

The objective function $z$ represents a linear combination of six criteria for each employee $e$. The actual paid working time $W'_e = \max\{W_e, 390\}$ is the main objective, and it is combined with the total time $T_e$ to reduce long unpaid periods for employees. The next sub-objectives reduce the passive ride time $ride_e$ and the number of tours changes $chg_e$, which is beneficial for both employees and efficient schedules. The last objective aims to reduce the number of split shifts $split_e$ as they are very unpopular. The weights were determined by previous work (Kletzander and Musliu 2020) based on preferences agreed by different stakeholders at Austrian bus companies and employee scheduling experts.

---

**Algorithm 1** Adaptive Large Neighbourhood Search

**Input:** $s_0$ (the initial solution), $k_0$ (initial destruction size)
**Output:** $s_{\text{bsf}}$
1: $k \leftarrow k_0$
2: $s_{\text{bsf}} \leftarrow s$
3: Initialise the weights $\boldsymbol{\rho}$
4: **while** time $< t_{\max}$ **do**
5:      Select destroy operator $d \in \Omega$ using $\boldsymbol{\rho}$
6:      $s' \leftarrow \text{CG}\left(d(s_{\text{bsf}}, k)\right)$
7:      **if** $z(s') < z(x_{\text{bsf}})$ **then**
8:          $s_{\text{bsf}} \leftarrow s'$
9:      **end if**
10:     Update weights $\boldsymbol{\rho}$ and sub-problem size $k$
11: **end while**
12: **return** $s_{\text{bsf}}$

---

## 3 Large Neighbourhood Search

The *Large Neighbourhood Search* (LNS) algorithm was introduced by Paul Shaw in 1998 (Shaw 1998). The main idea is to destroy part of a solution in order to obtain a sub-problem that is easy to solve optimally or at least close to optimality. Selecting the part to destroy is done by a set $\Omega$ of *destroy operators* (or *destroyers*), the operator to apply is chosen randomly proportional to a given weight vector $\boldsymbol{\rho}$. Solving the sub-problem is done by a *repair operator*, often an exact method. We accept the new solution $s'$ if $z(s') < z\left(s_{\text{bsf}}\right)$, where $z$ represents the objective function value (1) and $s_{\text{bsf}}$ is the best-so-far solution. Algorithm 1 shows the pseudo-code of the algorithm.

### 3.1 Destroy Operators

Since our repair mechanism can only produce complete shifts, the aim of the destroyers is to select a subset of employees $E' \subseteq E$ that is removed from the current solution. The size of the sub-problem $k = |E'|$ is given to the destroy operator. We propose three distinct ways to select $E'$:

**EU** Employees uniform: Select $k$ employees uniformly.

**EW** Employees weighted: $\left\lfloor \frac{k}{2} \right\rfloor$ of the employees are selected using their cost as weight, the others uniformly. This is motivated by the fact that employees with high cost have a higher potential to benefit from reoptimisation. The split is done since a combination of high-cost and low-cost shifts can have a better potential to balance the shifts in the sub-problem, e.g., by transferring some legs from the high-cost shift to an underutilised shift.

**TR** Tour remover: A tour is uniformly selected and all employees that share at least one leg of this tour are removed. This process is iterated until at least $k$ employees are removed. This operator is based on the idea of selecting employees that have something in common and therefore have a higher potential that useful recombinations of their shifts are possible, e.g., optimising when and where a bus is handed over from one driver to the other. Note that this operator might select more than $k$ employees because it removes all the employees who share a tour.

However, tours are usually not shared by too many employees since this incurs extra cost, so $|E'|$ does typically not exceed $k$ by much.

## 3.2 Repair Operators

Once a set of removed employees $E'$ is selected, the repair mechanism needs to solve the sub-instance that is created by using all legs $\ell$ assigned to any employee $e \in E'$ together with the common data for the whole instance. This sub-instance represents a complete instance of BDSP and can therefore be solved with any solution method of choice.

Since the Branch and Price approach (Kletzander, Musliu, and Van Hentenryck 2021) is the most powerful for small instances (it can provide an optimal solution for instances with 10 tours within seconds), it is the best fit for solving these sub-instances.

Branch and Price (Barnhart et al. 1998) works by splitting the problem into a master problem and a sub-problem. The master problem is set partitioning (Balas and Padberg 1976) which assumes that a set of shifts is already given including costs for each shift and tries to find the minimum cost subset of these shifts that covers each bus leg exactly once.

$$\text{minimise} \sum_{s \in \mathbf{S}} cost_s \cdot x_s \qquad (2)$$

$$\text{subject to} \sum_{s \in \mathbf{S}} cover_{s\ell} \cdot x_s = 1 \qquad \forall \ell \qquad (3)$$

$$x_s \in \{0, 1\} \qquad \forall s \qquad (4)$$

Here $x_s$ is the variable for the selection of shift $s$. The objective (2) minimises the total cost, Equation (3) states that each bus leg needs to be covered exactly once (using $cover_{s\ell} \in \{0, 1\}$ to indicate whether shift $s$ covers leg $\ell$), and Equation (4) states the integrality constraint.

The sub-problem is a Resource Constrained Shortest Path Problem (RCSPP) (Irnich and Desaulniers 2005) where each leg is represented by a node in an acyclic graph, and each possible shift corresponds to a path in this graph from a source node to a target node. Costs and constraints are represented by resources that are tracked for each path through the graph and need to adhere to certain limits. Duals from solving the relaxed master problem (no integrality constraint) are added for each node, and each resource-feasible path where the cost of the edges and nodes on the path minus the sum of all duals along the path is negative (negative reduced cost) have the potential to improve the solution of the master problem. The complex rules for each shift are modelled in this sub-problem, making it very challenging to solve. Therefore, several optimisations were necessary to solve it efficiently (Kletzander, Musliu, and Van Hentenryck 2021).

Master problem and sub-problem are repeatedly solved until no more path with negative reduced cost can be found. This part of the process is called Column Generation and results in the optimal solution for the relaxed master problem, however this result is usually fractional. Therefore, branching is done and Column Generation is repeated on a modified problem where some connections from the graph are removed. This branching process is repeated until all branches are closed or until timeout.

However, previous work (Kletzander, Musliu, and Van Hentenryck 2021) already shows that, for instances up to 60 tours, the results are very close to the optimum when only solving Column Generation on the root node and then solving the master problem with integrality constraint on the set of columns obtained during Column Generation. These solutions are often much faster, but achieve a gap of around 1% while the following branching process only closes this remaining gap very slowly.

Therefore, we propose to drop the aim of optimally solving the sub-instance with Branch and Price, and instead only use Column Generation on the root node to get very good solutions to the sub-instance very fast. In the evaluation, we compare using Column Generation (CG) with using full Branch and Price (BP).

Once the repair mechanism returns a solution consisting of employees $E^*$ that contain all bus legs from $E'$, the new solution for the full problem is provided by $(E \backslash E') \cup E^*$.

## 3.3 Sub-problem Size

An important parameter for large neighbourhood search is the size of the sub-problem. However, the appropriate size depends on the destroy and repair operators. In the case of our system, the destroy operators are easy and fast to apply, but the complexity of Branch and Price increases rapidly with the size. Even when just applying Column Generation, the size of the RCSPP in the sub-problem still leads to considerable increases in runtime.

Therefore, based on preliminary experiments, the smallest sub-problem size in use is $k = 5$. This size can still be solved in a few seconds, so it is fast enough, but it also leads to a high number of improvements, so it is large enough to allow meaningful changes of the solution. In the process of the search this size can be increased if too many iterations without improvement occur. This indicates that a larger size might be needed to escape local optima.

We use a maximum size of $k_{\max} = 20$ since runtime grows rapidly and for larger size too much time would be spent on each individual sub-problem. When running the algorithm, the size starts with an initial value of $k_0$, and is increased by 1 until reaching $k_{\max} = 20$ whenever the previous improvement was more than $n_{\max}$ iterations ago. As soon as an improvement is found, $k$ is set back to the initial value $k_0$.

## 3.4 Adaptivity

*Adaptive Large Neighbourhood Search* (ALNS) is an extension of LNS, where the weights $\boldsymbol{\rho}$ for selecting the operators are adapted dynamically based on their performance (Ropke and Pisinger 2006).

Our method takes into account the score and the time required by destroy operator $i$. At first, every component of the weight vector $\boldsymbol{\rho}$ is set to $\frac{1}{|\Omega|}$. The destroy operator is selected in a random way with weights $\boldsymbol{\rho}$ using the *roulette wheel principle*: the probability of choosing the $i$-th destroy

operator is defined as follows:

$$\mathbb{P}\left(i\text{-th operator is selected}\right) = \frac{\rho_i}{\sum_{j=1}^{|\Omega|} \rho_j} \qquad (5)$$

The selected destroy operator is then applied to the current solution $s$, which results in a sub-problem that is passed to the repair operator $r$. At iteration $n$, we update the weight $\rho_i^n$ of the $i$-th destroy operator using the following equation:

$$\rho_i^{n+1} = \lambda \rho_i^n + (1 - \lambda) \frac{\sum_{j=0}^{n} s_i^j}{\sum_{j=0}^{n} t_i^j} \qquad (6)$$

where $s_i^j = 1$ if the $i$-th operator has improved the best-known-solution at iteration $j$, else 0. The denominator is a sum of runtimes, so $t_i^j$ represents the time the $i$-th operator took for the entire process (destroying + repairing) at iteration $j$. If operator $i$ was not selected at iteration $j$, then $s_i^j = t_i^j = 0$. The real parameter $\lambda \in [0, 1]$ controls the sensitivity of the weights. A value of $\lambda$ close to 0 implies that the most recent results have a large influence while a value of 1 keeps the initial weights static.

As long as the denominator is 0, the value of the fraction is set to 0. In this case, $\rho_i^{n+1} = \lambda \rho_i^n$.

## 4 Evaluation

All executions were performed on a cluster with 11 nodes equipped with Ubuntu 22.04.2 LTS. Each node has two Intel Xeon E5-2650 v4 (max 2.20 GHz 12 physical cores, no hyperthreading). For each run, we set a memory limit of 4.267 GB and use one thread. The implementation is in Python, executed with PyPy 7.3.11. BP is implemented in Java, using OpenJDK 20, and CPLEX 22.11 for the master problem.

We evaluate the quality of a solution using the relative gap (GAP) compared to the best-known solution in the literature:

$$\text{GAP}(x) = \frac{z(x) - z(x_{\text{bks}})}{z(x_{\text{bks}})} \cdot 100, \qquad (7)$$

where $x$ is the solution and $x_{\text{bks}}$ is the best-known solution. A negative value of $\text{GAP}(x)$ means a new best-known solution has been found. We use the GAP to have a metric quality that does not scale with the instance dimension.

All experiments are performed with 1 h of wallclock time. Our method is non-deterministic. Therefore, we execute 10 runs for each instance. Random seeds for different runs are recorded for reproducibility.

### 4.1 Instances and Initial Solution

We use the publicly available sets of benchmark instances provided by previous work (Kletzander and Musliu 2020; Kletzander, Mazzoli, and Musliu 2022)[1]. There are 65 instances in 13 sizes, ranging from around 10 to around 250 tours. Note that the last 3 sizes have much larger size changes than the previous sizes.

The initial solutions were generated using a greedy construction method (Kletzander and Musliu 2020), assigning

---

[1]https://cdlab-artis.dbai.tuwien.ac.at/papers/sa-bds/

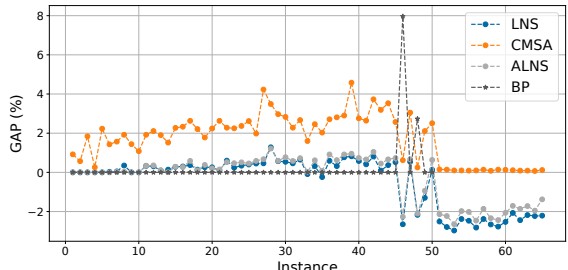

Figure 3: GAP for different methods

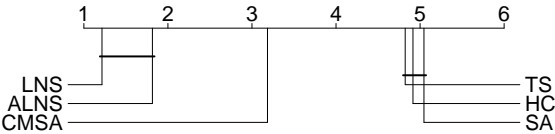

Figure 4: Critical difference plot for all the instances.

bus legs to the employee where the lowest additional cost is incurred, or to a new employee if this would incur an extra cost of at most 500 compared to the best existing employee assignment.

### 4.2 Results

We first present the final results, obtained with the best configurations using $k_0 = 10$, $n_{\max} = 50$, the repair operator CG, and only the destroyer TR for LNS, or the parameter $\lambda = \frac{2}{3}$ and all destroyers for ALNS. Details about the operator selection are then presented in the next subsection.

Table 2 shows the objective function values for LNS, ALNS, CMSA (Rosati et al. 2023) (same time budget) and BP (Kletzander, Musliu, and Van Hentenryck 2021). For each size there are 5 instances, each value in the table is the average of the 5 results per size due to space limitations. The full table is available as supplementary material.

BP is the state-of-the-art method for smaller instances. It optimally solves the first 10 instances (*), except for instance 7. For the smallest size it only takes few seconds. It has a known optimality gap of less than 1 % for all instances up to size 60 and provides the best-known solutions up to size 90 and 2 instances of size 100. However, it uses many resources. Results starting from size 60 use extra time on four threads after a full hour of BP to obtain an integer solution, up to an additional hour ($^\dagger$), while our approach is very close to these results while always keeping to 1 h and a single thread. Up to size 100, the average of LNS outperforms previous BP results on 5 instances, the minimum on 8 instances. For instances larger than size 100, BP fails with a memory error even when given 8 GB of RAM, and with larger memory allowance, it still produces solutions worse than the results from the construction heuristic, since the integer master problem is just becoming too large to solve in reasonable time.

The previous state-of-the-art incomplete method, CMSA, was able to outperform earlier meta-heuristics like Simulated Annealing and Tabu search. In comparison, we can

Table 2: LNS and ALNS with previous results from BP and CMSA. For each size there are 5 instances. For LNS, ALNS, and CMSA, we present the objective function values and standard deviation of ten runs. BP, being deterministic, does not require this information.

| | LNS | | | ALNS | | | BP | CMSA | | |
|---|---|---|---|---|---|---|---|---|---|---|
| size | average | min | std | average | min | std | value | average | min | std |
| 10 | **14 709.2** | **14 709.2** | 0.0 | **14 709.2** | **14 709.2** | 0.0 | **14 709.2**[*] | 14 867.4 | 14 879.7 | 14.2 |
| 20 | 30 322.1 | 30 297.2 | 23.3 | 30 298.8 | 30 294.6 | 7.2 | **30 294.8**[*] | 30 695.8 | 30 745.9 | 41.6 |
| 30 | 49 974.5 | 49 874.2 | 77.9 | 49 956.6 | 49 888.2 | 80.4 | **49 846.4** | 50 731.4 | 50 817.2 | 56.3 |
| 40 | 67 183.8 | 67 030.2 | 91.5 | 67 207.5 | 67 069.4 | 118.8 | **67 000.4** | 68 394.8 | 68 499.9 | 68.7 |
| 50 | 84 622.9 | 84 459.0 | 122.1 | 84 697.4 | 84 462.0 | 193.6 | **84 341.0** | 86 219.0 | 86 389.2 | 104.0 |
| 60 | **100 362.1** | **100 101.6** | 160.3 | 100 492.7 | 100 226.2 | 231.0 | 99 727.0[†] | 102 596.2 | 102 822.9 | 160.6 |
| 70 | **118 790.2** | **118 592.0** | 156.0 | 119 002.8 | 118 737.0 | 237.0 | 118 524.2[†] | 120 935.6 | 121 141.9 | 112.0 |
| 80 | **135 349.3** | **134 966.4** | 273.8 | 135 629.8 | 135 091.2 | 413.7 | 134 513.8[†] | 138 406.8 | 138 760.3 | 281.1 |
| 90 | **151 032.2** | **150 694.4** | 217.6 | 151 436.5 | 150 991.0 | 359.3 | 150 370.8[†] | 154 692.6 | 155 078.3 | 309.3 |
| 100 | **167 081.1** | **166 720.2** | 271.6 | 167 538.3 | 166 931.0 | 495.3 | 172 582.2[†] | 171 159.4 | 171 786.7 | 323.6 |
| 150 | **256 218.3** | **255 501.6** | 400.7 | 257 283.8 | 256 402.0 | 945.9 | | 263 079.2 | 263 387.7 | 216.9 |
| 200 | **339 313.7** | **338 459.2** | 545.5 | 340 824.4 | 339 526.2 | 1150.1 | | 348 608.6 | 349 017.0 | 221.5 |
| 250 | **428 963.7** | **428 170.6** | 554.9 | 431 238.5 | 429 657.8 | 1239.0 | | 438 811.4 | 439 234.5 | 297.4 |

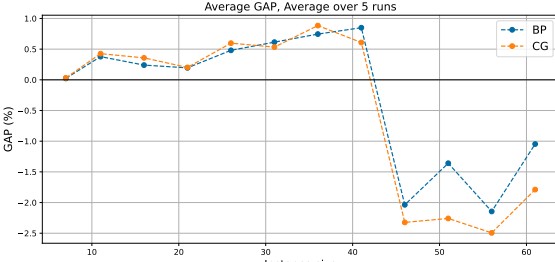

Figure 5: GAP for different repair operators

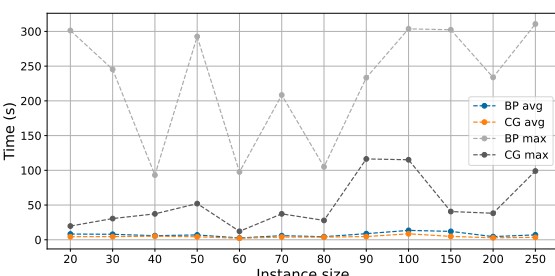

Figure 6: Runtime of different repair operators

now outperform CMSA on all 65 instances by 0.26 to 3.62 %, especially by at least 2.18 % on the instances of size 100 and above, showing the strength of our method. Overall, limited to a timeout of 1 hour on a single thread, we outperform all other methods on size 60 and above, and provide 23 new best-known solutions.

Figure 3 shows the average GAP for LNS, ALNS, BP, and CMSA. Points below the $x$-axis (i.e., instances with a negative GAP) show new best solutions.

**Statistical Significance.** We compared CMSA and LNS across all 65 instances using the Wilcoxon signed-rank test (Calvo and Santafé 2016). We used the `scipy` module (version 1.11.1) of Python. With a significance level of $\alpha = 0.05$, the test confirms that the $p$-value is smaller than $2.4 \times 10^{-16}$, showing a significant difference between the two algorithms.

Moreover, we performed statistical tests using the R script SCMAMP (Calvo 2021), also including Simulated Annealing and Hill Climbing (Kletzander and Musliu 2020), and Tabu Search (Kletzander, Mazzoli, and Musliu 2022). As described by Calvo (Calvo and Santafé 2016), we first applied the Friedman test to detect whether all algorithms perform the same, which is rejected with a $p$-value smaller than $2.4 \times 10^{-16}$. Then, we compared multiple algorithms using the Nemenyi *post-hoc* test (Calvo and Santafé 2016).

We graphically show the results on all instances with a *Critical Difference* (CD) plot in Figure 4. Each considered algorithm is placed on the horizontal axis according to its average ranking for the instances (lower is better). The performances of those algorithm variants below the critical difference threshold (0.94) are considered statistically equivalent. In the CD plot, this is remarked by a horizontal bold bar that joins different algorithms.

The results show that LNS performs better than ALNS, but not with significant difference, while both versions significantly outperform CMSA, which in turn significantly outperforms the previous meta-heuristics.

## 4.3 Evaluation of Algorithmic Components

To select our final parameters, we thoroughly analysed the impact of different algorithmic components on a subset of instances from the benchmark set. Since all algorithms show similar performance on instances of the same size, we chose one instance from each size, skipping the smallest size that can be solved to optimality with BP in seconds. Therefore,

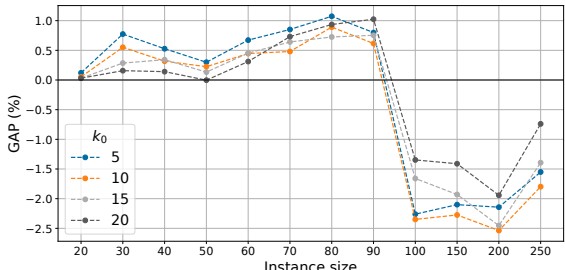

Figure 7: GAP for different values of $k_0$

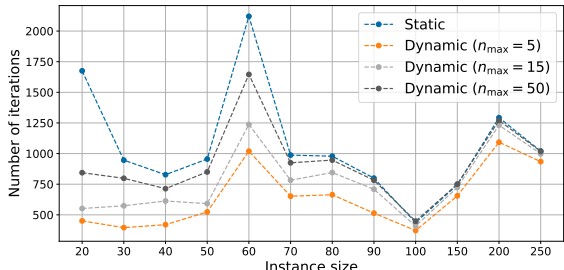

Figure 9: Number of iterations for different values of $n_{\max}$

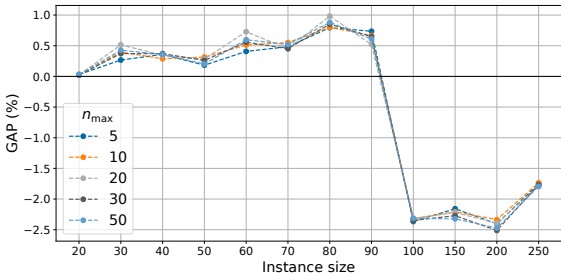

Figure 8: GAP for different values of $n_{\max}$

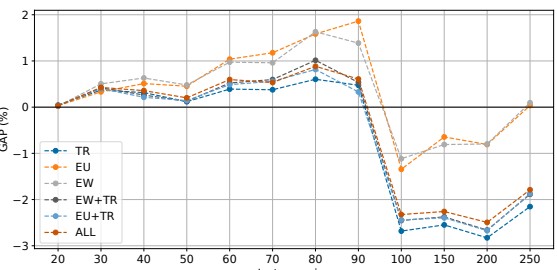

Figure 10: GAP for different subsets of destroyers

we used 12 instances in this part of the evaluation; each result is the average of 5 runs. It includes the following components:

1. The repair mechanism: BP or CG
2. The initial destruction size $k_0$
3. Max. number of iterations without improvement $n_{\max}$
4. The destroyer selection
5. The role of adaptivity

**Repair Operator Selection.** For fixed $k_0 = 10$, $n_{\max} = 50$, and equal selection of all destroyers, we compared BP and CG. Initial experiments showed that the performance of destroy and repair is rather independent from each other, which enables separate evaluation. Each repair operation has a maximum budget of $5\,\mathrm{min}$, but is expected to usually terminate much faster.

Results are shown in Figure 5. The performance is very similar for the smaller instances, but CG is clearly the better choice for larger instances, showing that the extra time used for repairing in BP is not justified. Figure 6 shows that BP usually takes longer than CG, but while the average time in both cases is under $10\,\mathrm{s}$, BP often reaches the time budget of $5\,\mathrm{min}$, while CG is always below $2\,\mathrm{min}$, showing a much better worst case behaviour.

**Initial Destruction Size.** There are several options regarding the destroyers, first we investigated the initial destruction size $k_0$, fixing all other parameters. The size remained constant, CG and all destroyers are used, and $\rho_i = 1/3$ without adaptivity.

We tested sizes $k_0 \in \{5, 10, 15, 20\}$. Figure 7 shows the results. While $k_0 = 20$ performs slightly better for smaller instances, it is outperformed on larger instances. Overall,

$k_0 = 10$ seems best for the large instances which are the main focus of this work, therefore we fix $k_0 = 10$.

**Number of Iterations Without Improvement.** Next, we investigated increasing the size $k$ every $n_{\max}$ iterations without improvement by 1, until reaching the upper bound $k_{\max} = 20$ or finding an improvement.

We tested $n_{\max} \in \{5, 10, 15, 20, 30, 50\}$, but found no significant difference among them, as shown in Figure 8. We decided to set $n_{\max} = 50$, since it still allows to increase the size when needed, but does not increase it very often. We tried starting with different values for $k_0$, but found similar results, the initial size is more important than the step.

Figure 9 shows the impact of $n_{\max}$ on the number of iterations (CG calls). Larger values of $n_{\max}$ imply less frequent size changes, so more iterations. Of note is that for the larger instances, the size barely changes, as improvements are frequently found even with the initial size until timeout.

**Destroyer Selection.** In order to understand the impact of the destroy operators, we tested all the 7 possible combinations of them. Figure 10 shows that TR has the biggest impact on the performance. Is shows the best results on its own, with very similar results using in in any other combination, while all combinations without TR show significantly worse performance, with higher divergence among larger instances.

The advantage of selecting employees that share tours is that the sub-problems are more likely to allow meaningful optimisations. This hypothesis is further backed by the success rate (percentage of iterations where the current solution could be improved) in Figure 11, which shows that in general for larger instances more improvements in sub-

Table 3: Parameters, their domains, and the chosen values.

| Parameter | Domain | Value | Description |
|---|---|---|---|
| $k_0$ | $\{5, 10, 15, 20\}$ | 10 | Initial Destruction Size |
| $n_{\max}$ | $\{5, 10, 15, 30, 50\}$ | 50 | Max number of iteration without improvements |
| $d$ | $\{\texttt{EU}, \texttt{TR}, \texttt{EW}, \texttt{EU} + \texttt{TR}, \texttt{EW} + \texttt{TR}, \texttt{TR} + \texttt{EU} + \texttt{EW}\}$ | $\texttt{tr}$ | Destroy operator(s) |
| $\lambda$ | $\{\frac{1}{3}, \frac{1}{2}, \frac{2}{3}\}$ | $\frac{2}{3}$ | Decay parameter |

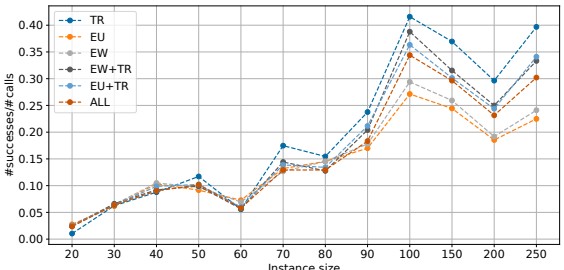

Figure 11: Success rate for different subsets of destroyers

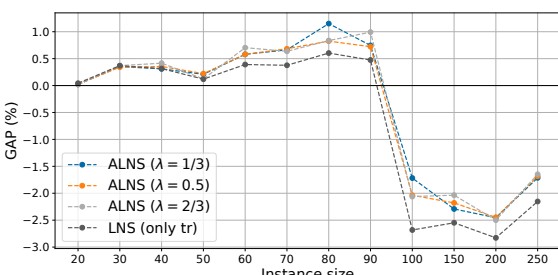

Figure 13: GAP for adaptive and static LNS

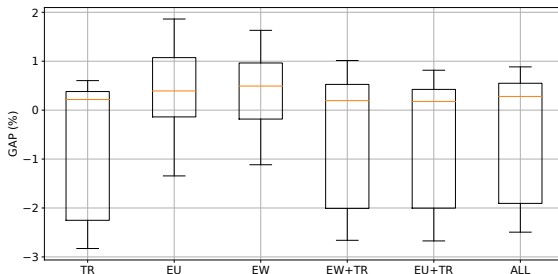

Figure 12: Box plot for different subsets of destroyers

problems can be found, but especially using just $\texttt{TR}$ has a higher success rate than any other set of destroyers. Additionally, Figure 12 shows that the Tour Remover operator is essential to obtain high-quality solutions (the $\texttt{EU}$ and $\texttt{EW}$ destroy operators alone are not enough).

While only using $\texttt{TR}$ is the best choice, we further investigated several non-uniform weight distributions with high weights for $\texttt{TR}$. Denoting the weights as $(\rho_{\texttt{EU}}, \rho_{\texttt{EW}}, \rho_{\texttt{TR}})$, we conducted experiments with $(5, 5, 90)$, $(10, 10, 80)$, and $(25, 25, 50)$. The first two where very similar to $\texttt{TR}$ only, while $(25, 25, 50)$ started to get slightly worse.

**Adaptivity.** To investigate the impact of adaptivity, we conducted experiments by changing the parameter $\lambda$ in (6), considering all three destroy operators. We tested three different values for $\lambda$: $\frac{1}{3}$, $\frac{1}{2}$, and $\frac{2}{3}$.

Figure 13 suggests that the adaptivity does not improve the average GAP with respect to the solely $\texttt{TR}$, and different values of $\lambda$ do not show significant difference.

## 5 Conclusions

This paper studied a complex version of the Bus Driver Scheduling Problem and proposed a new hybrid approach based on Large Neighbourhood Search to solve it. The results show that our algorithm outperforms all previous metaheuristics on the problem, and significantly improves all best-known solutions for larger instances as well as some mid-sized instances. Although exact Branch and Price remains the best method for smaller instances, our results come very close on mid-sized instances while using fewer computational resources, making it the best choice for a wide range of realistic instance sizes.

We further compared and evaluated several algorithmic components, including different repair options, several parameters for the destroy operator selection, and the role of adaptivity. Results show that Column Generation is more effective as a repair operator than full BP, and in particular the bus tour structure and the initial destruction size play a crucial role during the destruction phase.

Note that while this paper only addresses one set of rules, the methodology of using LNS with Column Generation as a repair operator is applicable to different kinds of rule sets or even different problems with a similar structure where individual shifts need to cover a set of tasks. Adaptations to different problems can be done by changing only the subproblem (RCSPP) used in Column Generation. Adapting a different set of rules to fit into the framework of the RCSPP can be challenging, but all other components are independent of the specific rules. This will be interesting to investigate in future work. Moreover, since this method provides very good results, applying LNS with Column Generation to other complex problems will be considered for future work.

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
