# OpenReview forum: "Investigating Large Neighbourhood Search for Bus Driver Scheduling"
_icaps-conference.org/ICAPS/2024/Conference — ICAPS 2024_

### Official Review · Reviewer_dBkc · 2024-01-17

**Significance And Importance:** 1
**Soundness:** 4
**Novelty:** 1
**Clarity:** 4
**Overall Evaluation:** 1
**Confidence:** 4

**Weaknesses:**

1: Minor weaknesses that are easily fixable.

**Contributions Of The Paper:**

This paper describes the application of an Adaptive Large Neighbourhood Search (ALNS) algorithm to a complex bus driver scheduling problem. The goal of the problem is to assign bus drivers to predetermined routes subject to a variety of constraints originating from collective agreements and European regulations. The paper describes an ALNS algorithm with three destroy operators and one repair operators (which comes from previous work of the same authors). A computational study evaluates the ALNS algorithm on instances previously published by the same authors and compares it against algorithms from the literature. It is shown how the ALNS algorithm outperforms existing methods by 0.26 to 3.62%. A series of experiments is conducted to understand the impact of different algorithm components and parameter settings on its performance.

**Ethical Considerations:**

(5) Excellent: The paper comprehensively addresses all of the applicable ethical considerations

**Nomination For Best Paper:**

No

**Questions For Authors:**

- The concept of a 'tour' is not clearly explained in Section 2. Does it correspond one-to-one to a specific vehicle?
- How did you define the probability update rule (6)?
- What is the reference value used to compute the GAP in Figure 3? Equation (7) refers to best-known solutions from the literature, but which results are these? Not from CMSA or BP?

**Reproducibility:**

5: Code and domains (whichever apply) are already publicly available

**Strengths Of The Paper:**

- New best results on a set of benchmark instances are obtained.
- The paper is well-written and structured in a clear manner.

**Weaknesses Of The Paper:**

- The paper does not introduce any novel algorithm components or problem elements. ALNS algorithms are well-known. The proposed destroy operators are rather standard and generic. The repair operator is more advanced, but was previously published by the same authors. The problem itself and the used benchmark instances were previously published by the same authors.
- The improvements over existing algorithms is very small (0.26 to 3.62%), but the authors show it is statistically significant.
- The evaluation is done by comparing the new algorithm only to methods previously published by the same authors.

---

> ### Author Rebuttal · Authors · 2024-01-27
>
> Novelty: There was still a considerable gap for large instances that are very relevant in the real world (e.g., see Comparison section in [1]) and out of reach for exact methods. One aspect of novelty lies in using CG within LNS, which is very limited in literature, and in this domain is very challenging due to the complex subproblem. We further propose destroy operators which are effective, and use domain structure. We also provide a detailed analysis of several aspects of LNS, including advantages of using CG vs. BP for the subproblem. The method is also of more general interest since the subproblem is applicable to different kinds of rule sets or even different problems with a similar structure where shifts need to cover a set of tasks. The setting from literature that we use is complex, used in the real-world, and provides a set of publicly available real-world-like benchmark instances which is rare in related work.
>
> Improvements: This is a real-world domain where the schedules of workers are one of the main cost components, and even small reductions of working hours result in significant cost savings. A large part of the objective is fixed (all bus legs need to be covered), leading to small improvements measured in percent, which translate to important cost saving without any loss in service quality.
>
> Comparison: We compare our method to all state-of-the-art results evaluated on the same benchmarks independent of authors. In the future we aim to extend our evaluation and comparison to methods evaluated on other problem variants.
>
> Q1: Yes, tours are IDs of the vehicles.
>
> Q2: We update weights considering the number of successes and the total time of its selections. A similar approach was used in a related crew scheduling domain [2]. Their score is simplified to the number improvements since we never accept worse solutions, and we use total time instead of calls due to potentially larger differences in runtime.
>
> Q3: We use the best-known solutions in the literature obtained by BP for instances up to size 90, and by CMSA for larger instances. and GAP to have a metric quality that does not scale with dimension. The GAP CMSA shows in Figure 3 for the larger instances is the gap between average and best CMSA values.
>
> [1] Kletzander and Musliu. Solving large real-life bus driver scheduling problems with complex break constraints. ICAPS 2020.
> [2] Carmo and Silv., An Adaptive Large Neighborhood Search Heuristic to Solve the Crew Scheduling Problem. 2019.

---

### Official Review · Reviewer_pYmm · 2024-01-19

**Significance And Importance:** 2
**Soundness:** 4
**Novelty:** 3
**Clarity:** 4
**Overall Evaluation:** 2
**Confidence:** 5

**Weaknesses:**

1: Minor weaknesses that are easily fixable.

**Contributions Of The Paper:**

This paper tackles a scheduling problem already known in the literature as the Bus Driver Scheduling Problem (BDSP). The BDSP is solved using a metaheuristic known as Large Neighborhood Search (LNS) which has proved very effective in tackling scheduling problems in the literature. The major contributions of the paper derive from the attempt to adapt the LNS schema to the BDSP, which basically translates into introducing the two building blocks of the procedure, the destroy and the repair operators (more specifically, a set of three destroy operators and basically two different repair operators). The authors evaluate the proposed approach against a sufficiently significant benchmark set of BDSP instances, producing interesting results. Moreover, a further LNS feature is investigated, by assessing the LNS's performance in the adaptive setting (ALNS), in which the destroy operators are selected dynamically during the optimization process.

**Ethical Considerations:**

(1) Not Applicable: The paper does not have any ethical considerations to address

**Nomination For Best Paper:**

No

**Questions For Authors:**

1) Section 3.4, page 5, 1st column, in the description of formula (6): how is the information about all the possible past improvements obtained by all the operators in the previous iterations kept?

2) Relatively to the ALNS: being ALNS adaptive, one might expect that both procedures will eventually achieve the same performance, as the weight \rho should eventually "learn" to select the TR destroyer as this is the most effective (i.e., the one that has best improved the past solutions). Can the authors elaborate on this?

**Reproducibility:**

5: Code and domains (whichever apply) are already publicly available

**Strengths Of The Paper:**

The paper is well structured and rather clear to read. The solution proposed by the authors to solve the BDSP demonstrates a rather good performance. The chosen Destroy and Repair operators are reasonably devised, and the difference in their potential performance sufficiently motivated and explained. From the strictly technical standpoint I do not see particular flaws.
The experimental evaluation provides sufficiently crisp results.

**Weaknesses Of The Paper:**

The paper presents no major weaknesses. Maybe the mean weakness is the very limited significance of the results obtained from the adaptivity-related analysis, which did not keep the promises one would have hoped for. Indeed, the ALNS approach returned very marginal improvements, if any, which can only be explained by the fact that the superiority of the TR destroy operator w.r.t. EW and EU revealed so high that no dynamic combination of the other approaches could keep up. However, my impression is that this aspect will require further investigation.

---

> ### Author Rebuttal · Authors · 2024-01-27
>
> Q1: At every iteration, for each destroyer $d$, the nominator represents the total number of successful calls, and the denominator stores the total time spent by $d$.
>
> Q2: We agree with the reviewer that over time, ALNS is expected to learn that TR performs better, and indeed, we see the selection probability of TR rising over time. However, especially for the larger instances, there are not enough iterations (around 1000) to fully learn this behaviour in the given runtime, and the improvement rates are higher for all destroyers. Therefore, the head start LNS with TR only explains the difference in results.

---

### Official Review · Reviewer_Mfqr · 2024-01-20

**Significance And Importance:** 2
**Soundness:** 4
**Novelty:** 2
**Clarity:** 3
**Confidence:** 4

**Weaknesses:**

-1: Major weaknesses requiring significant work to be addressed for the paper to be accepted.

**Contributions Of The Paper:**

This paper presents an ALNS application for the Bus Driver Scheduling Problem (BDSP), a real-world optimization problem in Austrian public transportation domain with practical constraints. The goal in this problem is to assign bus drivers to predetermined routes in a way that minimizes costs and maximizes employee satisfaction, while adhering to complex problem-specific regulations (e.g., working time regulations with shift splitting).
To address this problem, the paper applies a Large Neighbourhood Search (LNS) approach, incorporating several destroy operators and a repair strategy using Column Generation. The authors have analyzed the impact of the different operators, and outperform branch and price.

**Ethical Considerations:**

(5) Excellent: The paper comprehensively addresses all of the applicable ethical considerations

**Nomination For Best Paper:**

No

**Overall Evaluation:**

-1: (weak reject)

**Questions For Authors:**

- Why have you not tuned the weight values of the ALNS and its sensitivity parameter λ, which are crucial for its performance.
- Why have you decided to not use any acceptance criterium in LNS and ALNS (such as simulated annealing)?
- In Figure 6, why is the runtime of BP max larger for instances of a smaller size, assuming that the destroy severity is remained the same? If these are edge cases, why do you only include the average runtime here (maybe with some standard deviations)?

**Reproducibility:**

5: Code and domains (whichever apply) are already publicly available

**Strengths Of The Paper:**

- The major strength is its application to a real-world problem, with some very complex constraints that are present in real-world problems, yet are often ignored in research. This makes the work a very interesting read, and potentially having impact on the work shift planning of Austrian bus drivers.

**Weaknesses Of The Paper:**

- I am not sure about the novelty of the paper. The studied problem (bus driver scheduling) is very specific, inspired from real-world but taken from the literature. This specific problem has been solved by a (similar) group of researchers using different methods, from exact method, meta-heuristics, and various search based methods (tabu, local search, and iterated local search). The authors of this work apply another search based method (LNS and ALNS) with relatively generic operators to this problem, using random removal, a weight based removal and a tour removal operator. These operators all have been suggested before for generic routing problems. Given the problem is very specific and the proposed LNS is a commonly used approach for scheduling and routing, I think the impact of this paper is limited.
- As a paper submitted to the application track, I was hoping to see a good evaluation of the method in a realistic setting. However, the instances used are from the previous work and the properties (how close to real-world instances) were not described. The authors also didn’t  provide discussion on the impact of the work to the real-world problem.
-  On of the main contributions of the work is its ability to get faster results, using lower resources. However, a study with different search budgets is missing from the work (how do LNS/ALNS perform when the search budget is e.g., halved/doubled).
- The writing of the paper can be improved; especially of the problem formulation is unclear. Also, the parameter tuning of the different proposed methods should be explained before the results section, and the authors do not explain the CMSA benchmarking method in the paper.

---

> ### Author Rebuttal · Authors · 2024-01-27
>
> Novelty / Impact: The problem is applied in a real-world scheduling domain where even small reductions of working hours translate to significant cost savings. There was still a considerable gap for large instances which are very relevant in the real world (e.g., see Comparison section in [1]) and out of reach for exact methods. One aspect of novelty lies in using CG within LNS, which is very limited in literature, and in this domain very challenging due to the complex subproblem. We further propose destroy operators which are effective, and use domain structure. We also provide a detailed analysis of several aspects of LNS on our domain. The method is also of more general interest since the subproblem is applicable to different kinds of rule sets or even different problems with a similar structure where shifts need to cover a set of tasks.
>
> Real-world: While real instances are not available due to confidentiality, benchmarks were carefully crafted to be as close to reality as possible (see section Instances in [1]), including realistic distribution of demand throughout the day and idle times between bus legs. Regarding impact, please refer to the previous answer.
>
> Time budgets: We compared CMSA and LNS for 5, 15, and 30 min. In every case, LNS performed better. BP, starting from mid-sized instances, only gives a result at the end of its runtime. We only presented 1h due to space limitations and because this was used in the literature.
>
> Writing: We will improve these aspects.
>
> Q1: We tried different values for lambda, as shown in Results - Adaptivity, however, without significant difference. We also tried fixed rho = (5, 5, 90), (10, 10, 80), and (25, 25, 50). The first two were very similar to TR only, the last closer to the adaptive cases.
>
> Q2: We didn't use these because the improvement rates were good, we did not get stuck in local optima, especially on the larger instances which are of most interest.
>
> Q3: Maximum runtimes for BP can vary a lot depending on the branching steps. The plot conveys that the average runtime drops significantly (7.3 s for BP, 4.3 s for CG, std.dev: BP 3.4, CG 2.3), but especially edge cases get reduced a lot. This is important because aborting CG might result in higher gaps. Showing the edge cases supports that BP runs into timeout on several occasions, but CG can easily complete within the timeout.
>
> [1] Kletzander and Musliu. Solving large real-life bus driver scheduling problems with complex break constraints. ICAPS 2020.

---

### Meta-Review · Area_Chair_DUfv · 2024-02-06

**Recommendation:** Accept (Poster)
**Confidence:** 5

**Metareview:**

This paper presents a Large Neighborhood Search (LNS) approach to solving a real-world bus driver scheduling problem (BDSP) that attempts to optimize both operating costs and driver satisfaction, while satisfying complex constraints emanating from collective agreement rights and government regulations. The aim is an effective solution for large problem instances that are beyond the reach of exact methods. Several destroy and repair operators are proposed and configured to develop a baseline search procedure, and a mechanism for selecting them adaptively during the search is also explored. An experimental analysis is performed on a previously published set of benchmark problems, and new best-known results are obtained for all problems.

Strengths: The technique is applied to a real-world problem with several complicated constraints that are not typically considered in research settings. The benchmark problem set considered in the paper are derived from the real-world problem faced by the Austrian bus drivers and can potentially impact the effectiveness of future operations. The analysis rightly focuses on large problem instances that are outside of the reach of exact methods, and technique is shown to produce new best-known results (with confirmed statistical significance. The presentation is clear and provides sufficient background for the paper to be self-contained.

Weaknesses: The main weakness of the paper is its perceived lack of novelty. The innovation in the approach over previous work is not emphasized in the presentation and appears to be quite limited. Several of the solution components comprising the configured LNS have been previously published, and the mechanisms of LNS /ALNS technology are generally well known. What are the keys to application success here?

If the paper is accepted, please incorporate author(s) rebuttal comments made in response to reviewer questions about novelty and respond to other reviewer criticisms raised when preparing the final version of the paper.

**Ethical Considerations:**

(5) Excellent: The paper comprehensively addresses all of the applicable ethical considerations